# Lessons from a systematic tracing process aimed to reduce initial loss to follow-up (ILTFU) among people diagnosed with tuberculosis (TB) in Cape Town, South Africa

Nosivuyile Vanqa[1]*, Lario Viljoen[1], Graeme Hoddinott[1,2], Anneke Hesseling[1], Muhammad Osman[1,3], Sue-Ann Meehan[1]

1 Department of Paediatrics and Child Health, Faculty of Medicine and Health Sciences, Desmond Tutu TB Centre, Stellenbosch University, Tygerberg, South Africa, 2 Faculty of Medicine and Health, School of Public Health, The University of Sydney, Sydney, Australia, 3 Faculty of Education, Health and Human Sciences, School of Human Sciences, University of Greenwich, London, United Kingdom

* nvanqa@sun.ac.za, nosivuyile@gmail.com

## Abstract

### Background

South Africa is a high tuberculosis (TB) burdened country. People who are newly diagnosed with TB must link to a TB treatment facility and be registered in the electronic TB notification system for ongoing care. Delayed linkage to care increases the risk of disease progression, mortality, and ongoing TB transmission. We describe lessons from a systematic tracing process aimed to support linkage to care for people diagnosed with TB.

### Methods

The study used the Western Cape Provincial Health Data Centre (PHDC) to identify persons newly diagnosed with TB (January-December 2020) who were not recorded as linked to care after routine linking efforts, in one peri-urban health sub-district in Cape Town, South Africa. A systematic tracing process was followed, including visits to primary health care (PHC) facilities, and home visits for those with no evidence of linkage at PHC level. Descriptive statistics were used to analyse quantitative data. Lessons learned during the process were documented.

### Results

Within the PHDC, 406 persons diagnosed with TB had no evidence of being linked to TB care. Verification at PHC facilities found that 153/406 (38%) had linked to care. We traced 219 persons; of which107 (49%) could not be found. Overall, the PHDC showed 76% linkage among those traced and found and 72% among those not found. Lessons learned include the need for improved; (i) record keeping enabling

**Data availability statement:** The data set file is available from the Figshare database (accession number(s) CC BY 4.0. https://doi.org/10.6084/m9.figshare.27179805

**Funding:** Professor Anneke C Hesseling is the PI of the study and received funding from the Bill and Melinda Gates Foundation (BMGF) https://www.gatesfoundation.org/. Foundation (BMGF), investment ID: INV-007130. The contents are the responsibility of the authors and do not necessarily reflect the views of the BMGF. The funders had no role in study design, data collection, and analysis, decision to publish, or preparation of the manuscript. GH receives financial assistance from the European Union (Grant no. DCI-PANAF/2020/420-028), through the African Research Initiative for Scientific Excellence (ARISE), pilot programme. ARISE is implemented by the African Academy of Sciences with support from the European Commission and the African Union Commission. The contents of this document are the sole responsibility of the author(s) and can under no circumstances be regarded as reflecting the position of the European Union, the African Academy of Sciences, and the African Union Commission. ACH is financially supported by the South African National Research Foundation (NRF) through a South African Research Chairs Initiative (SARChI). The financial assistance of the NRF towards this research is hereby acknowledged. Opinions expressed, and conclusions arrived at, are those of the authors and are not necessarily to be attributed to the NRF.

**Competing interests:** The authors have declared that no competing interests exist.

the allocation of resources to patients who are truly lost to follow up, (ii) communication to improve patient understanding of timely treatment initiation and (iii) interpersonal relationships to encourage trust.

## Conclusion

The systematic tracing process was useful to understand the complexities around delayed linkage to care. To reduce ILTFU, we recommend, improving accuracy and timely recording of TB data, updating patient contact details regularly and strengthening interpersonal relations and communication between patients and healthcare workers.

## Background

The Tuberculosis (TB) epidemic is a leading public health crisis [1], TB replaced COVID-19, as the leading cause of death by a single infectious agent causing an estimated 1.25 million deaths globally in 2023 [1]. To achieve a TB-free society, it is vital to detect TB early and start TB treatment [2]. At the UN high level meeting on TB in 2023, there was commitment to scale up efforts to close the gaps along the pathway of care, including diagnosis and treatment gaps [1]. The World Health Organization estimated that only 75% of people who developed TB in 2023 were included in a TB treatment registration system [1].This leads to underestimates of case notifications and true TB burden. People who are diagnosed with TB but who are not included in the official TB reporting system are referred to as initial loss to follow-up (ILTFU).

South Africa is a high TB burden setting, with an estimated TB incidence of 270,000 cases in 2023, of which 47,000 (17%) were either not diagnosed or not entered into a TB reporting system [1]. ILTFU in South Africa has been reported as 12% among persons with drug-susceptible (DS) TB [3], with higher rates (between 16% and 37%) among those with drug-resistant (DR) TB [4]. ILTFU varies across provinces (12% to 32%), by place of diagnosis, with ILTFU among persons diagnosed with TB in hospitals much higher (21%−64%) compared to those diagnosed at primary healthcare facilities (12%−24%) [5], and by age, with high rates of ILTFU among children with DR-TB [6]. ILTFU poses significant health threats; people with TB who have not initiated treatment are at a higher risk of mortality and morbidity [7] and add to the onward transmission of Mycobacterium tuberculosis [8]. It is vital that we address ILTFU and find efficient ways to link people with TB to care and treatment.

Both systemic and personal challenges result in people with TB delaying linkage to care and treatment. Persons with TB reported poor interpersonal relationships and a lack of communication with healthcare providers, leading to a misunderstanding about their diagnosis and required treatment initiation [9]. People newly diagnosed with TB may also fear TB treatment and its' side effects, while some may be in denial of their positive TB test result [10]. Health facilities

typically do not update patient contact details at each health visit, which may result in delays in tracing and time to treatment initiation [11,12]. ILTFU remains a major gap in the health system that must be addressed in a holistic manner.

Previous South African studies have reported various interventions to reduce ILTFU; including the use of mHealth [13], using a dedicated referral system from hospital to PHC level for children [14], community tracing strategies using technology [15], and counselling and conditional cash transfers [16]. Other studies have suggested and the use of community outreach workers as a resource to reduce ILTFU [17,18]. It is unclear if any of these interventions have been systematically implemented as part of routine services. Embedded within the routine health services, this study used the routine data available for TB diagnosis and treatment initiation and e aimed to evaluate the yield, effort required, and document lessons learned from a systematic follow-up process to find people who were ILTFU.

## Methods

### Study design

We report on work nested within a larger project that aimed to reduce ILTFU among people with TB in South Africa (LINKEDin) by testing the implementation of health system strengthening activities in the hospital and at primary healthcare (PHC) [5]. In the LINKEDin study, people eligible to link to a PHC facility, but with no evidence of linkage received an alert-and-response people management intervention; a short message service (SMS), a phone call and/or a referral for a community-based healthcare worker (CHW) to do a home visit to support linkage to care (routine in the South African context).

This was a prospective observational cohort study that implemented a systematic follow-up of people newly diagnosed with TB, who had no electronic record of linkage to care after having received the alert-and-response people management interventions.

### Setting

This study took place in the Khayelitsha sub-district in the Cape Town Metro district, Western Cape Province (WCP) of South Africa. Khayelitsha is a peri-urban low-resourced area with a mix of formal and densely populated informal housing. In the province, TB investigations, diagnosis and treatment initiation takes place at both primary and tertiary health care facilities. TB services are decentralised, and TB registration and treatment for drug-susceptible TB (DS-TB) and drug-resistant TB (DR-TB) are provided at PHC facilities and specialized TB hospitals. The TB program is nurse-driven, with clinical oversight provided by medical officers. TB services in Khayelitsha are provided at 13 public PHC facilities and one district hospital. In the WCP, hospitals are not TB registration facilities, so people diagnosed in hospitals must register at a PHC facility or a specialized TB hospital for ongoing TB care and treatment.

The South African TB program is supported by CHWs, whose roles include tracing and linking people to care and supporting them to remain in care [19–21]. In Cape Town, this cadre of staff is employed by non-governmental organizations (NGOs) and seconded to the Department of Health (DoH). Typically, they work within defined catchment areas around health facilities. Working in pairs, they leave the health facility each day, with a list of people that they need to find and/or visit for specific services [22].

In Cape Town, the reporting of all persons diagnosed with TB occur at designated TB treatment sites which includes PHC facilities and specialized TB hospitals, where TB treatment is provided to persons with TB. People diagnosed in general hospitals are referred to the PHC facilities for recording and continuation of care. People are typically first registered in a paper-based register and thereafter entered into the electronic TB treatment register at the facility using the unique identification number "personal number/folder number". This includes capturing the existing personal information as per referral letter, including the details on the clinic file and or updated information where necessary for tracing and follow-up.

The electronic TB register is managed at a sub-district level and aggregated at the district level. In the WCP, the Provincial Health Data Centre (PHDC) consolidates personal-level clinical data across public health services to support care [23]. The PHDC integrates multiple routine health data sources from public health services (e.g., personal information, laboratory, pharmacy use, TB registers, HIV care) into single patient records, enabling health workers to monitor and manage people across disease areas and health facilities in the province [23].

## Study population and sampling

Our activities consisted of pre-determined and iterative steps between the study team, the PHDC team, and PHC facility staff. We used the PHDC to identify all people diagnosed with TB (bacteriological or clinical diagnosis) and who had no electronic evidence of linkage to a TB treatment facility after the alert-and-response people management interventions.

The systematic process was implemented by trained research personnel. A data clerk (part-time) had access to the PHDC (as part of the LINKEDin study) and generated a list of persons who had been diagnosed with TB three weeks earlier and had no evidence of having linked to care. Two fieldworkers (full-time) worked together and carried out the tracing, and a supervisor (part-time) oversaw the process. The field workers used a designated vehicle fitted with an electronic Global Positioning System (GPS) navigation tool.

The field workers used the list generated by the data clerk and visited the relevant health facilities to check if there was evidence of these persons being registered at PHC facility level, prior to tracing them in the community. If people were registered at a PHC facility, in either a paper-based or electronic TB register, this information was fed back to the PHDC (Fig 1).

When field workers found persons in the community, they verified their identity and asked them about their recent TB test and diagnosis, and their linkage status. If these individuals reported to have already linked to a TB treatment facility

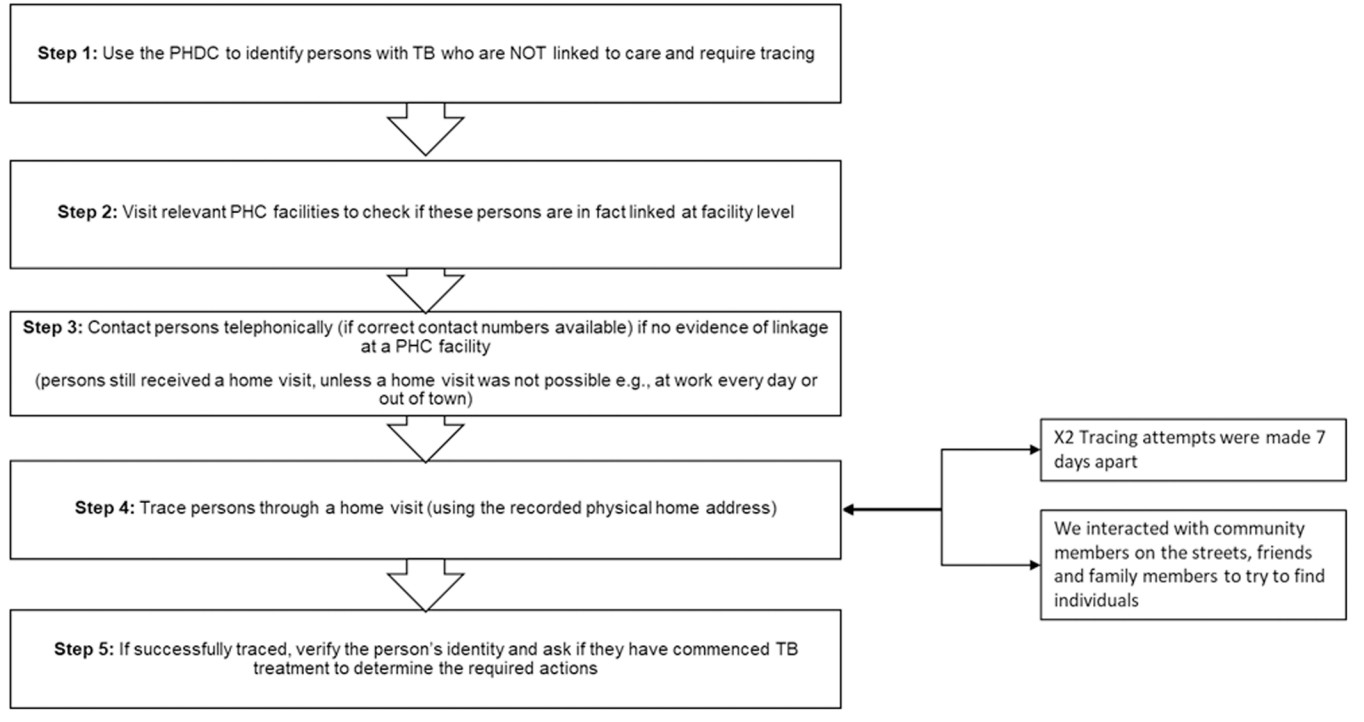

**Fig 1. The step-by-step tracing process.**

for TB care prior to the study team's visit, we verified this by reviewing their TB treatment card. Some persons could not be found at home (at work every day or out of town) but were contactable telephonically and a similar verification process was applied to them. Persons who had not linked to care at the time of the home visit or a call were referred to the PHC facility of their choice. During the team's interaction with these individuals, the importance of linkage to care, and the health risks associated with untreated TB were discussed. All tracing attempts occurred between 3 and 6 weeks after the lists were generated.

The study team interacted with relevant personnel at the PHDC on an ongoing basis, to inform and update them about changes in personal information and linkage status, including those confirmed as deceased by family members (a copy of a death certificate was submitted as supporting documentation to the PHDC).

### Data collection processes

Process data were collected and updated on Microsoft Excel spreadsheets. Tracing data were collected using the Research Electronic Data Capture (REDCap) system, a secure, web-based software for research studies [24]. During tracing activities, the study team systematically documented all efforts made to find and link people to TB care. This data extract included all persons with TB diagnosed between 01/01/2020 and 31/12/2020, irrespective of whether we were able to find them or not through our tracing process. To ensure adequate time for people to link to care, we obtained data extract from the PHDC on 30/04/2021 to determine the linkage status. Data verification and cleaning was done, and the final data set completed on the 31/05/2021. During the period of data collection, the reflective process involved both structured and informal approaches. Fortnightly team meetings and open post-data collection debriefs between the research assistants, socio-behavioural scientists and project leads, which allowed teams to share experiences and identify challenges. After data collection, the study team with our health services partners engaged in multiple policy-translation meetings. This was an iterative process that lasted for several months and resulted in consensus on key recommendations.

### Data analysis

We used descriptive statistics to analyse the data. We plotted our systematic process and calculated numbers and proportions for those already linked at PHC facilities, prior to tracing (not requiring home-based tracing), and those who required home-based tracing, and tracing outcomes. We used Pearson's chi square test to determine if there were any statistically significant differences in demographic and clinical characteristics between those who linked and had not linked to care. Lastly, we reflected on the activities and lessons learned from the systematic tracing processes.

### Ethical considerations

This study was nested within the LINKEDin study, which had approval from the Health Research Ethics Committee of Stellenbosch University (N18/07/069), the Western Cape Department of Health (NHRD ref: WC_201808_034), and the City of Cape Town Health Directorate (study ID number 8053). There was a waiver of informed consent to access routine health data for persons with TB. We conducted the study according to the guiding principles within the Declaration of Helsinki.

### Results

A total of 831 people were diagnosed with TB during the study period. Of these, 427 people diagnosed with TB had no evidence in the PHDC of having linked to a TB treatment facility, and 307 were captured on the REDCap data collection tool. Of these, 21/427 (5%) were excluded from the analysis because they were subsequently found not to have TB (based on subsequent TB test results showing a negative TB culture result). This data was updated in the PHDC. Of the remaining 406 persons with TB, 153/406 (38%) did not require home-based tracing, as they were confirmed to be in the TB treatment register at a PHC facility; 117/153 (77%) were recorded and captured in the facility's electronic TB register, while 36/153 (23%) were recorded in the facility's paper-based TB register.

### Tracing outcomes

Of the 253/406 (62%) persons with TB who required home-based tracing, 34/253 (13%) were excluded as they either only had a physical address outside of Khayelitsha sub-district or had no recorded address and no usable contact number in the routine health records. We attempted to first telephone and then do a home visit to trace the remaining 219 people with TB. Only 11 of these people had a useable and correct contact telephone number, with most of the numbers constantly going on automated response to leave a voice recorded message.

Of the 219 people with TB that we attempted to trace, 112/219 (51%) were found. Of those found, 55/112 (49%) showed us their TB treatment cards, confirming that they had already linked to care prior to the home visit (between the time spent confirming at the PHC facility and the tracing activities). The reasons for these persons not reflecting in the PHDC as linked, included individuals i) who were captured using a different name or unique identifier across data sources, ii) not captured in any TB register, or iii) whose TB treatment information was incorrectly captured, e.g., captured as receiving treatment preventative therapy (TPT) and not TB treatment. Of those not linked to care, 49/57(86%) were referred to care, 3/57 (5%) were confirmed as deceased by a family member, and 5/57 (9%) reported that they had not had a TB test, nor had they recently received a positive TB diagnosis.

There were 107 people with TB that we could not find. The majority (77%) could not be found either because we were unable to locate the home address provided in the routine records or the person was not at the recorded address despite two home-based tracing attempts on separate days (Fig 2).

### Linkage outcomes

The majority (37/49;76%) of people with TB that we traced, found, and referred to care were linked to a TB treatment facility. Of the 107 people that required home-based tracing but could not be found, 77/107 (72%), were later confirmed to be linked to care through the PHDC post-intervention. True loss to follow up was 30/107 (28%) among those not found and 12/49 (24%) among those found and referred to care (Fig 3).

### Characteristics of people with TB who were traced and found vs not found

Of the total number of people with TB who were traced and found (including those who linked prior to the home visit), more than half (57%) were between the ages of 15 and 35, 57/112 (51%) were diagnosed in a hospital, 92/112 (82%) were linked to care, and 8/112 (7%) died (see Table 1). People who could not be found were older (60% were 36 years and older), a high proportion (67/107; 63%) were diagnosed in a hospital, fewer linked to care (77/107; 72%), and 1/107 (1%) had died. From the 107 people we could not find, 30/107 (28%) were true lost to follow-up, of whom 1/30 (0.3%) died. Of those we traced, found and referred for treatment, 12/49 (24%) were true loss to follow-up, of which 3/12 (17%) died (Fig 3).

## Discussion

In this prospective observational study, we implemented a systematic tracing process to find 426 persons diagnosed with TB in one health sub-district of Cape Town. These were people who had no electronic evidence recorded in the PHDC, of having linked to care at a TB treatment facility. Using dedicated resources and following an intensive process, systematically implemented, we found that almost 40% of these persons had linked to care and were in a TB register at a PHC facility. This information was not evident in the PHDC. We found discrepancies between paper-based and electronic TB registers, typically due to poor data capturing, and incorrect and outdated personal details on the electronic systems. Other studies done in South Africa have assessed the quality of TB data and found that the data on paper-based registers did not correspond with data in the electronic registers (personal and TB information) [25–27]. Having an accurate electronic TB register, captured in 'real time' is vital, not only to truly reflect the work done within the TB program, but

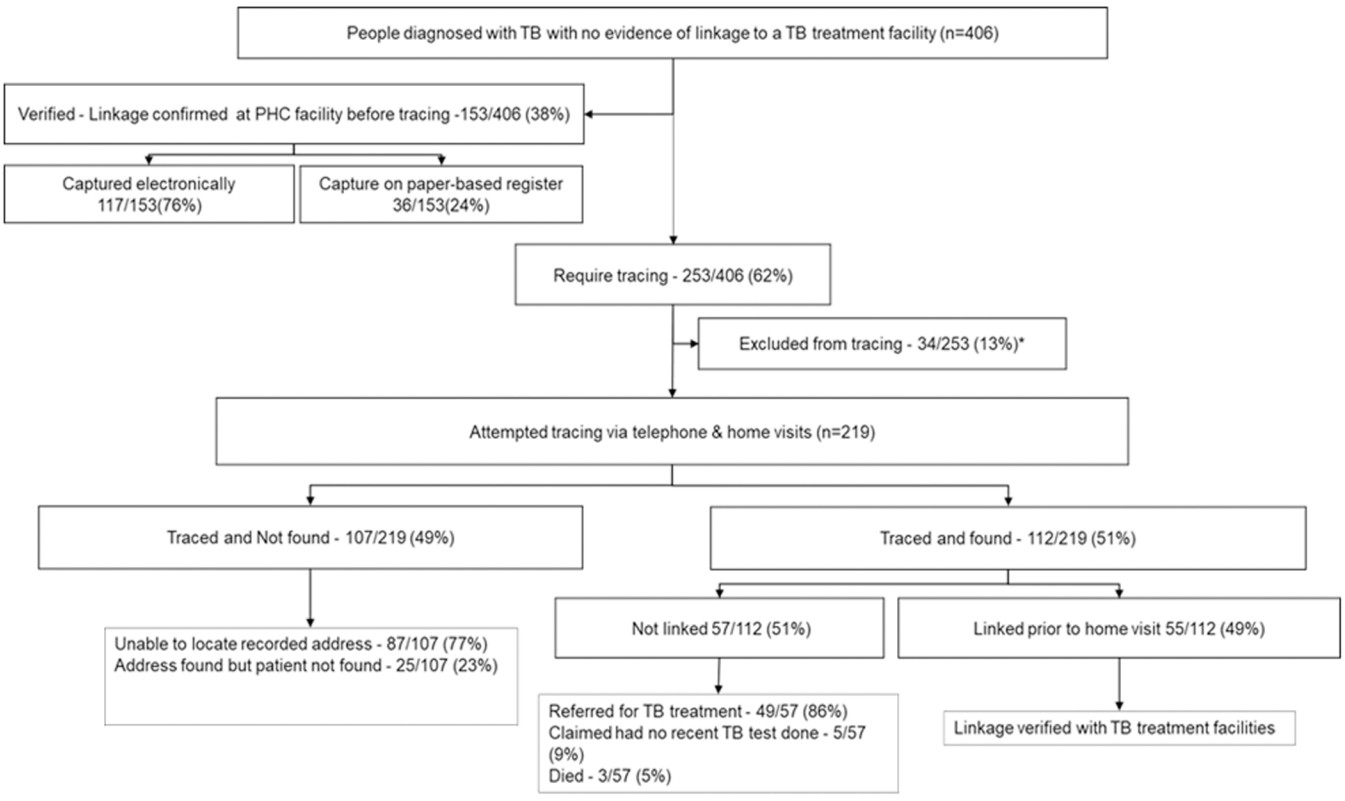

**Fig 2. Systematic tracing activity outcomes.** *Excluded from tracing due to: The address outside the study area/Absent or Incomplete address/No usable contact number.*

more importantly to reduce the resources invested in tracing people at home, who are already in care. Resources can be directed toward those who truly require additional assistance to link to care. We recommend that ongoing training for staff is prioritised to improve the accuracy and timely recording of TB data.

Of those we traced and found at home, almost half had linked to care in the time between us checking the records at the health facility and us visiting them at home. This is a similar finding to other studies done in similar contexts in South Africa [28,29]. Some people may just need additional time to link. Interventions to alert and identify people who have experienced a delay in linkage may be sufficient as a reminder to assist these people to link and would be more efficient than a home visit [5]. However, if people's contact details are not correctly recorded in the routine health records, then this will not be possible. In our study, we were only able to successfully telephonically contact 11/209 (5%) persons we needed to trace. A previous study in the same setting, only found 36% of people with TB in hospitals had the same contact number from what was captured in their hospital file records [12], with people lost to follow-up perhaps representing an especially socio-economically marginal group with greater life instability. The relatively low number of eligible people with contactable phone numbers even in a setting with relatively broad access to mobile phones may represent an ongoing challenge to high coverage of ILTFU interventions especially to the most socio-economically marginalised. There is an urgent need to update peoples' contact details at every healthcare visit [30]. We recommend that healthcare workers routinely check, and update patient contact details at every health visit and include landmarks where physical addresses are difficult to determine.

Anecdotally, field workers reported that people with TB who were found and referred for linkage had expressed they had not been sure of what to do after their TB test, hence the delay in returning to a PHC facility for treatment and care. Some people

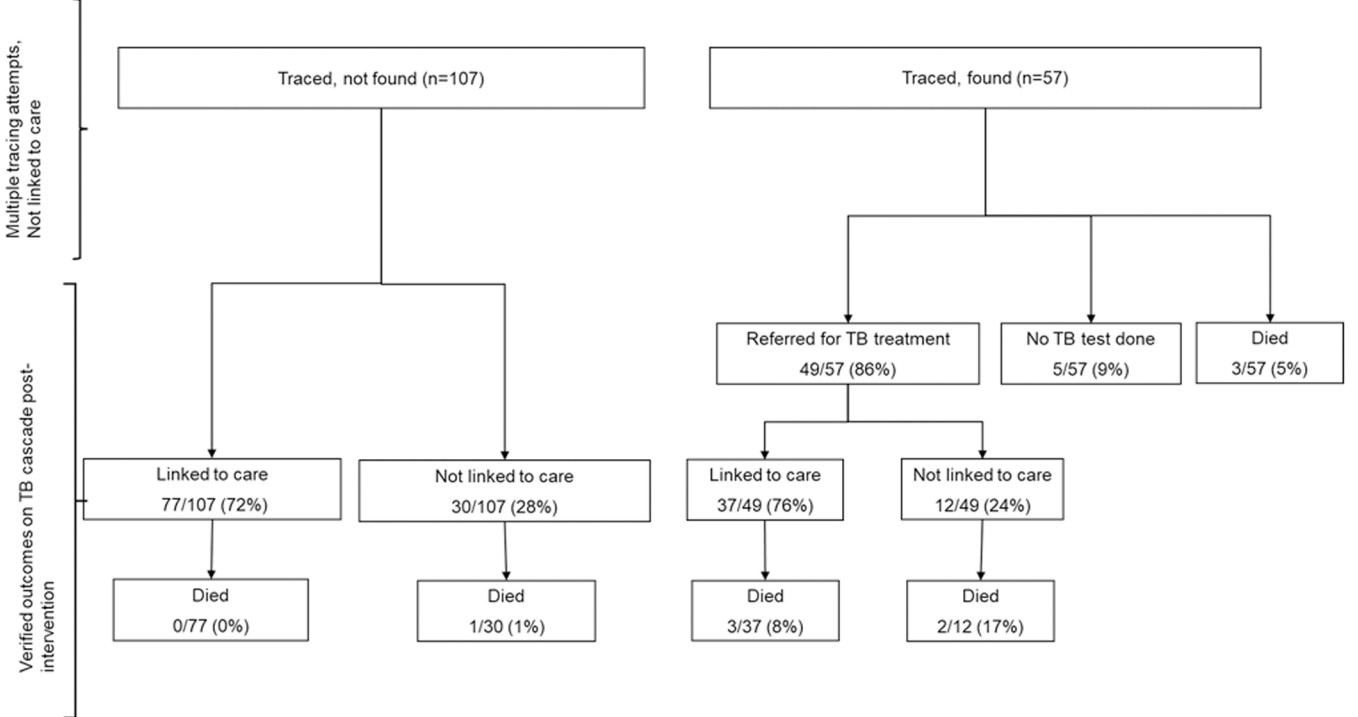

**Fig 3. Linkage to care outcomes for persons traced who were found and needed referral vs not found.**

claimed that they were informed that their test result was negative and were not aware of a positive TB test result (this could be due to a positive TB culture result after 6 weeks). This finding indicates the need for healthcare workers to spend more time explaining key information to individuals about what tests have been done, what to expect, and what the test results mean. A previous study highlighted the need for improved communication between healthcare providers and persons who are being tested for TB [9]. We recommend strengthening communication between healthcare workers and patients during the referral process to improve patient understanding around their TB diagnosis and the importance of initiating treatment without delay.

More than half of those we traced and could not find were diagnosed in hospitals. A study done in the same setting found that ILTFU among people diagnosed in hospitals is higher than among people diagnosed in PHC facilities, and most deaths occurred among people who were diagnosed in hospitals [7]. Being diagnosed in a hospital and needing to link to care at a PHC facility adds complexity to linkage for these individuals, as they may not have accessed PHC facilities previously. Other studies have also found poor continuity of care for people discharged from hospital [31,32] and may not know where to access care at PHC facilities. This highlights the importance and need for good communication between hospital staff and persons diagnosed with TB in hospital, to improve linkage for these persons.

People-centered approaches to facilitate linkage should be explored. Studies have shown that interactively engaging people in the hospital prior to their discharge has supported linkage to care [12,33]. Additionally, integrated counselling with the family members of those with TB empowers everyone with the knowledge that will ease fear and address concerns related to managing disease and an opportunity to alleviate stigma [34,35]. Structured counselling is an important component of quality TB services, it enhances knowledge to everyone and offers an opportunity to address concepts about the disease, treatment, transmission, and side effects [36]. Improved communication and coordinated referral systems are vital to address ILTFU among persons with TB. We recommend strengthening interpersonal relationships between healthcare workers and patients to build rapport and encourage trust and transparency.

**Table 1. Demographic and TB characteristics of persons with TB traced (n=219), comparing those found vs not found.**

| | ILTFU people found. N=112 (51%) | ILTFU people not found. N=107 (49%) | p-value |
|---|---|---|---|
| **Sex** | | | |
| Male | 61 (54%) | 56 (52%) | 0.752 |
| Female | 51 (46%) | 51 (48%) | |
| **Age** | | | |
| < 15 | 7 (6%) | 5 (5%) | 0.018 |
| 15-25 | 24 (21%) | 14 (13%) | |
| 26-35 | 40 (36%) | 24 (22%) | |
| 36-45 | 19 (17%) | 30 (28%) | |
| 46+ | 22 (20%) | 34 (32%) | |
| **Linked to TB care** | | | |
| Yes* | 92 (82%) | 77 (72%) | 0.073 |
| No | 20 (18%) | 30 (28%) | |
| **Died** | | | |
| Yes | 8 (7%) | 1 (1%) | 0.021 |
| No | 104 (93%) | 106 (99%) | |
| **Place of diagnosis** | | | |
| Hospital | 57 (51%) | 67 (63%) | 0.08 |
| PHC | 55 (49%) | 40 (37%) | |
| **Previous TB** | | | |
| Yes | 26 (23%) | 21 (20%) | 0.52 |
| No | 86 (77%) | 86 (80%) | |

*Includes people who linked to care between time spent confirming at the PHC facility and the tracing activities

*p-value – difference between groups tested for significance at 0.05 using Pearson's chi square test

Using a systematic tracing process with all necessary resources dedicated to our process, we were only able to find half of the individuals that we tried to trace. It remains challenging to physically locate persons in the community. The challenges we encountered included i) the person having moved out of the area, ii) the person not being known at the recorded address (indicative of having an incorrect home address on record) iii) no one at home on both tracing attempts or iv) the home address was in an informal settlement with no street name or number, so unable to locate. Other studies in South Africa have also found that it is difficult to find home addresses located in informal settlements [11,26,37]. A cohort study in Uganda showed that 43% of people with TB were lost to follow-up and 26% of these people could not be located or had moved from the district [38].

A major strength of this study is the use of a systemic tracing process with dedicated resources that enabled us to trace people across the whole sub-district. We were not restricted to one health facility catchment area, as is the current practice by the community healthcare workers who are responsible for tracing persons with TB in the community. Our systematic tracing process also included an iterative component that allowed us to reflect and communicate with data analysts at the PHDC regarding what we were finding in the community. Our feedback allowed the PHDC to update their data. While we acknowledge that person-related factors also play a role in linkage [9,10,39,40], this study highlights key aspects within the TB program that can be targeted to improve linkage to TB, by improving the data capturing quality for accurate personal information which will be linked with their unique identifiers to strengthen linkage to care, real-time data

availability; improved communication between healthcare workers and persons with TB and updating contact details at every healthcare visit.

The processes implemented were resource-intensive and time-consuming with limited yield. We would not propose that our systematic tracing process be implemented routinely, as the resources used to find half the persons needed to be traced do not seem to be an efficient use of resources. A costing analysis of this process is currently underway. However, the lessons learned and recommendations suggested are important and can be practically implemented within a real-world setting.

## Conclusion

Implementing a systematic tracing process was useful for understanding many of the underlying complexities around delayed linkage to care. Time and resources employed in finding these people can be minimised by training data clerks to improve the accuracy and timely recording of TB data and healthcare staff to improve communication with newly diagnosed people with TB. While we were able to find and link additional people to care, the proportion linked to care overall was similar for those we found and those we did not find. Our systematic tracing process showed that accurate data captured in 'real time' at health facilities will improve record keeping and allow for resources to be focused on persons with TB who are truly lost to follow-up and require home-based tracing.

## Acknowledgments

We acknowledge the support we received for this study from the healthcare staff and management working within the Khayelitsha health sub-district. We also acknowledge our close collaboration with persons at the Provincial Health Data Center (PHDC), especially Mariette Smith for her invaluable support and engagement. Our sincere appreciation to all persons with TB who engaged with the study tracing team.

## Author contributions

**Conceptualization:** Nosivuyile Vanqa, Graeme Hoddinott, Anneke Hesseling, Muhammad Osman, Sue-Ann Meehan.

**Data curation:** Nosivuyile Vanqa, Muhammad Osman.

**Formal analysis:** Nosivuyile Vanqa, Lario Viljoen, Muhammad Osman, Sue-Ann Meehan.

**Funding acquisition:** Anneke Hesseling.

**Investigation:** Nosivuyile Vanqa.

**Methodology:** Nosivuyile Vanqa, Lario Viljoen, Graeme Hoddinott, Anneke Hesseling, Muhammad Osman, Sue-Ann Meehan.

**Supervision:** Sue-Ann Meehan.

**Validation:** Nosivuyile Vanqa, Lario Viljoen, Muhammad Osman, Sue-Ann Meehan.

**Visualization:** Nosivuyile Vanqa, Lario Viljoen, Graeme Hoddinott, Muhammad Osman, Sue-Ann Meehan.

**Writing – original draft:** Nosivuyile Vanqa.

**Writing – review & editing:** Lario Viljoen, Graeme Hoddinott, Anneke Hesseling, Muhammad Osman, Sue-Ann Meehan.

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
