## [Decision Letter · Decision Letter 0]

5 Dec 2024

PONE-D-24-44791Lessons from a systematic tracing process aimed to reduce initial loss to follow-up (ILTFU) among people diagnosed with tuberculosis (TB) in Cape Town, South AfricaPLOS ONE

Dear Dr. Vanqa,

Thank you for submitting your manuscript to PLOS ONE. After careful consideration, we feel that it has merit but does not fully meet PLOS ONE’s publication criteria as it currently stands. Therefore, we invite you to submit a revised version of the manuscript that addresses the points raised during the review process.

We look forward to receiving your revised manuscript.

Kind regards,

Stephen Michael Graham, FRACP, PhD

Academic Editor

PLOS ONE

Journal requirements: When submitting your revision, we need you to address these additional requirements. 1. Please ensure that your manuscript meets PLOS ONE's style requirements, including those for file naming. The PLOS ONE style templates can be found at https://journals.plos.org/plosone/s/file?id=wjVg/PLOSOne_formatting_sample_main_body.pdf and https://journals.plos.org/plosone/s/file?id=ba62/PLOSOne_formatting_sample_title_authors_affiliations.pdf 2. One of the noted authors is a group or consortium [LINKEDin study team]. In addition to naming the author group, please list the individual authors and affiliations within this group in the acknowledgments section of your manuscript. Please also indicate clearly a lead author for this group along with a contact email address. 3. Please note that your Data Availability Statement is currently missing [the repository name and/or the DOI/accession number of each dataset OR a direct link to access each database]. If your manuscript is accepted for publication, you will be asked to provide these details on a very short timeline. We therefore suggest that you provide this information now, though we will not hold up the peer review process if you are unable.

Additional Editor Comments:

No additional comments.

Reviewers' comments:

Reviewer's Responses to Questions

**Comments to the Author**

1. Is the manuscript technically sound, and do the data support the conclusions?

Reviewer #1: Yes

Reviewer #2: Yes

2. Has the statistical analysis been performed appropriately and rigorously? 

Reviewer #1: N/A

Reviewer #2: Yes

3. Have the authors made all data underlying the findings in their manuscript fully available?

Reviewer #1: Yes

Reviewer #2: Yes

4. Is the manuscript presented in an intelligible fashion and written in standard English?

Reviewer #1: Yes

Reviewer #2: Yes

5. Review Comments to the Author

Reviewer #1: This article describes lessons learned from a systematic tracing process aimed to support linkage to care for people diagnosed with TB in one peri-urban health sub-district in Cape Town, South Africa. The work appears to have been conducted well and the manuscript is overall clear and well written.

No major issues to address.

Minor issues to address:

Abstract

• The last sentence of results isn’t clear standalone – see comment on related results section.

• The whole abstract is 340 words and was truncated in the review system. Instructions are that abstracts should not exceed 300 words - https://journals.plos.org/plosone/s/submission-guidelines

Introduction

• No comments

Methods

• Line 79/80 with reference 6 – it is unclear what this refers to as the reference gives only a year and volume which is 12 editions of the journal with multiple articles.

• Line 159 refers to a final extract ‘4 months after the end of the systematic tracing process’ but the abstract refers to ‘linkage outcomes five months after the end of the process.’

Results

• While not a key result, for context it would be helpful to know what proportion the 427 identified constitute of all people in the PHDC records for this period.

• Line 217-218 ‘Of the 107 people that required home-based tracing but could not be found, 77/107 (72%), linked to care (see Figure 3).’ This sentence perplexed me, as did the related sentence in the abstract (line 49). Figure 3 finally clarified that the way you knew their status was through verification in PHDC post-intervention. Please make this clear in the text.

• The section and table (1) on Characteristics of people with TB who were traced and found vs not found could be strengthened by some simple statistical comparisons of the proportions reported.

Discussion

• line 289 spelling error – patient-centred

• line 291 spelling error – supported

• line 208 spelling error – lost to follow-up

References

• Reference 6 and 12 are incomplete/insufficient information.

Reviewer #2: Thank you for the opportunity to review this manuscript articulating the lessons learnt when addressing a critical gap in the TB care cascade ie ILTFU, in Cape Town, South Africa

Overall, the paper is well written and provides great insights into the work aimed at strengthening linkage to care and treatment for persons with TB in Cape Town – high TB burden setting.

Please find a few comments below that could help clarify your findings and conclusions:

Main comments:

• A line or two under “Setting” to explaining the current system (if any) to link individual records from diagnosis to treatment and follow up, would be helpful eg is there a unique identifier system in use and how is that working or not working? Given that one of the key findings and lessons learnt from this study is the need to improve the capturing of individual records accurately especially the individual contact details, it is worth understanding the existing system that could be strengthened and how

• Similarly Line 318-321 – I would suggest to consider adding improved linkage of individual person records eg through a unique identifier or similar, to the proposed efforts to strengthen linkage to TB care

• I suggest highlighting what the “true” ILTFU was after all the efforts done to contact PHC facilities and persons with TB (telephone and home visits)

• Line 236 (Table 1) – comparison between 2 groups ie “ILTFU found” and “ILTFU not found”

o I am not a biostatistician but I think some analytical stats would be helpful to understand if there are differences between these two samples

Other comments:

• Line 59: “The Tuberculosis (TB) epidemic is a leading health crisis [1] with 10.6 million people diagnosed with TB in 2022; 23% of whom were in Africa”

o This should read “…an estimated 10.6 million…”

• Line 72: “aimed to evaluate the yield, effort required and lessons learned from a systematic follow up process to find people who were ILTFU “

o I suggest adding ‘document’ to read “… and document lessons learned…”

• Line 181-183 “Of these, 21/427 (5%) were excluded from the analysis because they were subsequently found not to have TB (based on subsequent TB test results showing a negative TB culture result).”

o 5% is a lot to have been excluded on the basis of a subsequent culture negative test. Is there data from other settings in South Africa and how does that compare with your study? Are these persons initially diagnosed with TB and referred to PHC for treatment? How are these persons excluded from the final TB notification data under the routine TB program?

• Line 193-194 Only 11 (out of 219) eligible for tracing has a contactable phone number. That is another major finding given a mobile phone is more accessible now and individuals from informal settlements are likely to have a phone (number) rather than a physical address. Is this due to inaccurate recording of the phone numbers or persons with TB deliberately gave wrong numbers? If the latter, then there is need to understand the reasons why and address them.

• Line 221-222: Figure 3: Linkage to care outcomes for persons traced who were found and needed referred vs not found.

o Probably a typo in the title. Did you mean “needed referral”?

• Line 241-244 – sentence too long. Consider revising it

• Line 264-265: “In our study, we were only able to successfully contact 11/209 (5%) persons we needed to trace”.

o I suggest adding that you were able to successfully contact by telephone otherwise can be misconstrued as the only ones you managed to reach through tracing

• Line 265-267 “A previous study in the same setting found 64% of TB patients in hospitals had an updated contact number from what was captured in their patient records [5].”

o Any reasons for this big difference between your study ie 5% vs 64% the other study in the same setting? I think this is worth discussing

• Line 289 – typo, should read “patient-centered…”

• The manuscript uses “patients” and “people with TB”. I suggest using the latter and for consistency

6. PLOS authors have the option to publish the peer review history of their article (what does this mean? ). If published, this will include your full peer review and any attached files.

Reviewer #1: No

Reviewer #2: **Yes: ** Dr Kudakwashe Chani

---

## [Author Response · Author response to Decision Letter 0]

20 Jan 2025

Dear Editor

Thank you for the opportunity to revise our paper. We attended to the recommendations and provided detailed feedback to each comment on the table below.

• Noted, thank you.

2. One of the noted authors is a group or consortium [LINKEDin study team]. In addition to naming the author group, please list the individual authors and affiliations within this group in the acknowledgments section of your manuscript. Please also indicate clearly a lead author for this group along with a contact email address.

• We have listed all the authors who are part of the manuscript and removed the statement that related to a group. The authors of this manuscript are limited to the people listed.

3. Please note that your Data Availability Statement is currently missing [the repository name and/or the DOI/accession number of each dataset OR a direct link to access each database]. If your manuscript is accepted for publication, you will be asked to provide these details on a very short timeline. We therefore suggest that you provide this information now, though we will not hold up the peer review process if you are unable.

• We have amended the data availability statement to read: “The data set file is available from the Figshare database (accession number(s) CC BY 4.0. “https://doi.org/10.6084/m9.figshare.27179805”

• We have reviewed and updated the references list.

Thank you to both reviewers for raising these important points, we appreciate the time taken to review and give advice on how to improve our manuscript.

Reviewer 1: Comments

1. Abstract: The last sentence of results isn’t clear standalone – see comment on related results section. The whole abstract is 340 words and was truncated in the review system. Instructions are that abstracts should not exceed 300 words.

• Noted, we have revised the sentence.

• We cut accordingly, the word count is now 300 words.

2. Methods: Line 79/80 with reference 6 – it is unclear what this refers to as the reference gives only a year and volume which is 12 editions of the journal with multiple articles.

• Thank you, we have updated this reference to: https://doi.org/10.1093/ofid/ofad648

3. Line 159 refers to a final extract ‘4 months after the end of the systematic tracing process’ but the abstract refers to ‘linkage outcomes five months after the end of the process.’

• Yes, the final data extract from the PHDC was done 4 months later. Thereafter, we verified and cleaned the data set, with the final outcomes concluded on the 5th month. We have revised this for clarity.

4. Results: While not a key result, for context it would be helpful to know what proportion the 427 identified constitute of all people in the PHDC records for this period.

• The PHDC is organized in disease cascades e.g. TB (which we utilized), HIV, pregnancy etc. Researchers obtain approval to access the relevant cascade. We cannot provide the total number of people recorded in the PHDC during this period. There were 831 persons recorded in the TB cascade (had a TB diagnosis) during this period.

5. Line 217-218 ‘Of the 107 people that required home-based tracing but could not be found, 77/107 (72%), linked to care (see Figure 3).’ This sentence perplexed me, as did the related sentence in the abstract (line 49). Figure 3 finally clarified that the way you knew their status was through verification in PHDC post-intervention. Please make this clear in the text.

• Thank you for the comment, we have clarified this in line 217-218 to ‘Of the 107 people that required home-based tracing but could not be found, 77/107 (72%), were later confirmed to be linked to care through the PHDC post-intervention (see Figure 3).

6. The section and table (1) on Characteristics of people with TB who were traced and found vs not found could be strengthened by some simple statistical comparisons of the proportions reported.

• Thank you for this suggestion. We have added a column in the table to show the p-value difference between groups tested for significance at 0.05.

7. Discussion: Line 289 spelling error – patient-centred.

• The error is corrected, thank you.

8. line 291 spelling error – supported.

• We corrected the error, thank you.

9. line 208 spelling error – lost to follow-up.

• We corrected the error, thank you.

10) References: Reference 6 and 12 are incomplete/insufficient information.

• We have updated reference 6 and checked reference 12, which appears complete.

Reviewer 2: Main Comments

1. A line or two under “Setting” to explaining the current system (if any) to link individual records from diagnosis to treatment and follow up, would be helpful eg is there a unique identifier system in use and how is that working or not working? Given that one of the key findings and lessons learnt from this study is the need to improve the capturing of individual records accurately especially the individual contact details, it is worth understanding the existing system that could be strengthened and how.

• We have added more information from line 108-116.

2. Similarly Line 318-321 – I would suggest to consider adding improved linkage of individual person records eg through a unique identifier or similar, to the proposed efforts to strengthen linkage to TB care.

• We have revised this sentence line337-342 “this study highlights key aspects within the TB program that can be targeted to improve linkage to TB, by improving the data capturing quality for accurate personal information which will be linked with their unique identifiers to strengthen linkage to care, real-time data availability; improved communication between healthcare workers and persons with TB and updating contact details at every healthcare visit”.

3. I suggest highlighting what the “true” ILTFU was after all the efforts done to contact PHC facilities and persons with TB (telephone and home visits)

• Thank you, we have added this on line 229-231.

3. Line 236 (Table 1) – comparison between 2 groups ie “ILTFU found” and “ILTFU not found”

I am not a biostatistician but I think some analytical stats would be helpful to understand if there are differences between these two samples.

• We compared the two groups with categorical variables using a chi square test and added a column in the table to show the p-value difference between groups tested for significance at 0.05.

Other Comments

1. Line 59: “The Tuberculosis (TB) epidemic is a leading health crisis [1] with 10.6 million people diagnosed with TB in 2022; 23% of whom were in Africa” This should read “…an estimated 10.6 million…”

• We corrected this, thank you.

2. Line 72: “aimed to evaluate the yield, effort required and lessons learned from a systematic follow up process to find people who were ILTFU “I suggest adding ‘document’ to read “… and document lessons learned…”

• Thank you for the suggestion which we have used.

3. Line 181-183 “Of these, 21/427 (5%) were excluded from the analysis because they were subsequently found not to have TB (based on subsequent TB test results showing a negative TB culture result).”

5% is a lot to have been excluded on the basis of a subsequent culture negative test. Is there data from other settings in South Africa and how does that compare with your study?

• Concern on Xpert false positivity in SA has been related to previous TB treatment history. In a study in the same city (Cape Town), as high as 1 in 7 Xpert-positive retreatment patients were culture negative and potentially false positive (https://academic.oup.com/cid/article/62/8/995/2462404). While this is an interesting track it is beyond the scope of this project. False positives need to be explored in terms of previous TB, timing since treatment, exposure and other factors but this is not the focus of this submissions

Are these persons initially diagnosed with TB and referred to PHC for treatment?

• Yes, they initially were recorded as having a positive TB test. A referral to a PHC facility would happen if they were tested at a hospital and discharged, but would have a return date to collect their results if tested at a PHC facility.

How are these persons excluded from the final TB notification data under the routine TB program?

• Thank you, this is an important point. There are a number of reasons for their removal which can include an error in the diagnosis or a clinical decision. Our study was not designed to further investigate the changes in the TB cascade but limited improving linkage to TB care.

4. Line 193-194 Only 11 (out of 219) eligible for tracing has a contactable phone number. That is another major finding given a mobile phone is more accessible now and individuals from informal settlements are likely to have a phone (number) rather than a physical address. Is this due to inaccurate recording of the phone numbers or persons with TB deliberately gave wrong numbers? If the latter, then there is need to understand the reasons why and address them.

• Poor data capturing is another contributing factor to this problem as described in our findings. Our study is not able to comment beyond this since we did not collect data on why there are so few patients with contactable phone numbers.

• We have highlighted this point in the discussion line 285-289: “The relatively low number of eligible people with contactable phone numbers even in a setting with relatively broad access to mobile phones may represent an ongoing challenge to high coverage of ILTFU interventions especially to the most socio-economically marginalised.”

5. Line 221-222: Figure 3: Linkage to care outcomes for persons traced who were found and needed referred vs not found. Probably a typo in the title. Did you mean “needed referral”?

• Yes, and thank you we have revised as suggested.

6. Line 241-244 – sentence too long. Consider revising it.

• Revised, thank you.

7. Line 264-265: “In our study, we were only able to successfully contact 11/209 (5%) persons we needed to trace”. I suggest adding that you were able to successfully contact by telephone otherwise can be misconstrued as the only ones you managed to reach through tracing.

• This is correct, we have revised the sentence.

8. Line 265-267 “A previous study in the same setting found 64% of TB patients in hospitals had an updated contact number from what was captured in their patient records [5].” Any reasons for this big difference between your study ie 5% vs 64% the other study in the same setting? I think this is worth discussing.

• We have revised the sentence in line 281-285 as “A previous study in the same setting, only 36% of TB patients in hospitals had the same contact number from what was captured in their patient records [5], with people lost to follow-up perhaps representing an especially socio-economically marginal group with greater life instability”.

9. Line 289 – typo, should read “patient-centered…”

• We corrected the error, thank you.

10) The manuscript uses “patients” and “people with TB”. I suggest using the latter and for consistency.

We have corrected this and mainly u

---

## [Decision Letter · Decision Letter 1]

9 Feb 2025

PONE-D-24-44791R1Lessons from a systematic tracing process aimed to reduce initial loss to follow-up (ILTFU) among people diagnosed with tuberculosis (TB) in Cape Town, South AfricaPLOS ONE

Dear Dr. Vanqa,

Thank you for submitting your manuscript to PLOS ONE. After careful consideration, we feel that it has merit but does not fully meet PLOS ONE’s publication criteria as it currently stands. Therefore, we invite you to submit a revised version of the manuscript that addresses the points raised during the review process.

The manuscript deserves a significant revision of the 'Background." The two-paragraph Background does not rationalize the study. The background is not sufficiently rooted in the existing literature on contact tracing and pre-treatment lost-to-follow-up persons with tuberculosis. A more meaningful introduction must answer these basic questions: What did we study? Why was it important to study the phenomenon? What do we know from the existing scholarship? How has the country of study addressed the concerns through its policies and programs? What research gaps is the study contributing to fill? The authors would realize that the Discussion section of the manuscript would benefit from further revision when they revise the Background along the suggested lines. It would be best for the authors to address these deficiencies to get the paper to a publishable level.Furthermore, while the paper's title focuses on lessons from a systematic tracing process, the Results section of the abstract does not indicate any lessons. This is a fundamental omission! The authors need to revise the Results section of the abstract to include essential statistics and lessons learned.Address the additional comments of the reviewers below.

We look forward to receiving your revised manuscript.

Kind regards,

Daniel Chukwuemeka Ogbuabor, Ph.D., M.D.

Academic Editor

PLOS ONE

**Additional Editor Comments:**

The manuscript deserves a significant revision of the 'Background." The two-paragraph Background does not rationalize the study. The background is not sufficiently rooted in the existing literature on contact tracing and pre-treatment lost-to-follow-up persons with tuberculosis. A more meaningful introduction must answer these basic questions: What did we study? Why was it important to study the phenomenon? What do we know from the existing scholarship? How has the country of study addressed the concerns through its policies and programs? What research gaps is the study contributing to fill? The authors would realize that the Discussion section of the manuscript would benefit from further revision when they revise the Background along the suggested lines. It would be best for the authors to address these deficiencies to get the paper to a publishable level.

Furthermore, while the paper's title focuses on lessons from a systematic tracing process, the Results section of the abstract does not indicate any lessons. This is a fundamental omission! The authors need to revise the Results section of the abstract to include essential statistics and lessons learned.

Address the further comments of the reviewers.

Reviewers' comments:

Reviewer's Responses to Questions

**Comments to the Author**

1. If the authors have adequately addressed your comments raised in a previous round of review and you feel that this manuscript is now acceptable for publication, you may indicate that here to bypass the “Comments to the Author” section, enter your conflict of interest statement in the “Confidential to Editor” section, and submit your "Accept" recommendation.

Reviewer #1: All comments have been addressed

Reviewer #2: All comments have been addressed

2. Is the manuscript technically sound, and do the data support the conclusions?

Reviewer #1: Yes

Reviewer #2: Yes

3. Has the statistical analysis been performed appropriately and rigorously? 

Reviewer #1: No

Reviewer #2: Yes

4. Have the authors made all data underlying the findings in their manuscript fully available?

Reviewer #1: Yes

Reviewer #2: (No Response)

5. Is the manuscript presented in an intelligible fashion and written in standard English?

Reviewer #1: Yes

Reviewer #2: Yes

6. Review Comments to the Author

Reviewer #1: The authors have adequately responded to the initial review. I have a few small comments in response that I suggest are addressed before the manuscript is considered final.

1. In response to item 3. The changes in abstract methods about timing (4 or 5 months) still lack clarity. If diagnoses were included up until 31/12/2020, but tracing started only 3 weeks after a person was not found to be linked in PHDC, then 31/05/2021 isn’t really 5 months after the ‘process’ concluded as tracing for the people last diagnosed would have started only in January. The sentence in methods (line 165) is improved. Consider changing ‘process concluded’ in the abstract – e.g. five months after the inclusion period?

2. Linked to the issue of period of intervention/study, duration of attempt to trace people was not clear. Figure 1 shows 2 attempts 7 days apart, and tracing started only 3 weeks after diagnosis, but was there a limit such that tracing attempts would only occur within 1-2 months after the lists were generated? A sentence at the end of the paragraph ending line 152 would be helpful, if word-count allows.

3. In response to item 6. Statistics have been added – the test name is ‘Pearson’s’ and it should be listed in the analysis section of the methods. The final p-value in Table 1 looked very surprising for these proportions and I think you will find it is 0.52.

4. Minor errors or word changes to consider:

a. Line 49-51: Accurate data captured in ‘real time’ at PHC facilities will improve records and the resources to can be focused on truly lost to follow-up people.

b. Line 131 remove unnecessary comma and I suggest change ‘ago’ to ‘earlier’: generated a list of persons who had been diagnosed with TB three weeks earlier

c. Line 224 add ‘were’: The majority (37/49;76%) of people with TB that we traced, found, and referred to care were linked to a TB treatment facility

d. Line 227 add missing closing bracket: True loss to follow up was 30/107 (28%) among those not found and 12/49 (24%) among those found and referred to care (see Figure 3).

e. Line 241-242 add a space between semi-colon and % figures in brackets.

f. Line 245: I suggest you reference Figure 3 at end of the sentence.

g. Reference 12 is not incomplete but it has ‘The’ immediately after the doi, which seems to be an error and should be deleted.

Reviewer #2: Comments

Thank you to the authors for making effort to address all my comments. Below are just minor comments (including typo corrections) to help improve the flow.

Lines 49-51: it appears there could be a missing word before “…the resources…”

Accurate data captured in ‘real time’ at 50 PHC facilities will improve records and the resources to be focused on truly lost to 51 follow-up people.

Lines 56-62

Suggest using the latest (2024) report

Lines 110-113 – suggest rewording

People are typically first registered in a paper-based register and thereafter into the electronic TB treatment register at the facility, using the unique identification number “personal number/folder number”.

Lines 164 correct typo, ‘c’ missing in care

Lines 242-245:

check calculation or denominator. 1/30 is 0.3%.

Review last part of the sentence ie “…those found and referred to care with (17%) of death rate.”

7. PLOS authors have the option to publish the peer review history of their article (what does this mean? ). If published, this will include your full peer review and any attached files.

**Do you want your identity to be public for this peer review?** For information about this choice, including consent withdrawal, please see our Privacy Policy .

Reviewer #1: No

Reviewer #2: **Yes: ** Kudakwashe Chani

---

## [Author Response · Author response to Decision Letter 1]

25 Mar 2025

Editor comments

1) The manuscript deserves a significant revision of the 'Background." The two-paragraph Background does not rationalize the study. The background is not sufficiently rooted in the existing literature on contact tracing and pre-treatment lost-to-follow-up persons with tuberculosis. A more meaningful introduction must answer these basic questions: What did we study? Why was it important to study the phenomenon? What do we know from the existing scholarship? How has the country of study addressed the concerns through its policies and programs? What research gaps is the study contributing to fill? The authors would realize that the Discussion section of the manuscript would benefit from further revision when they revise the Background along the suggested lines. It would be best for the authors to address these deficiencies to get the paper to a publishable level.

• The background has been completely revised. The discussion has been revised as required and in relation to the revisions in the background. The abstract has been revised to align with the revisions in the body of the manuscript.

2) Furthermore, while the paper's title focuses on lessons from a systematic tracing process, the Results section of the abstract does not indicate any lessons. This is a fundamental omission! The authors need to revise the Results section of the abstract to include essential statistics and lessons learned.

• We have substantially revised the abstract and added lessons learned into the results section of the abstract.

Reviewer 1: Comments

1) In response to item 3. The changes in abstract methods about timing (4 or 5 months) still lack clarity. If diagnoses were included up until 31/12/2020, but tracing started only 3 weeks after a person was not found to be linked in PHDC, then 31/05/2021 isn’t really 5 months after the ‘process’ concluded as tracing for the people last diagnosed would have started only in January. The sentence in methods (line 165) is improved. Consider changing ‘process concluded’ in the abstract – e.g. five months after the inclusion period?

• The abstract has been substantively revised and this sentence has been deleted to prevent unnecessary confusion.

2) Linked to the issue of period of intervention/study, duration of attempt to trace people was not clear. Figure 1 shows 2 attempts 7 days apart, and tracing started only 3 weeks after diagnosis, but was there a limit such that tracing attempts would only occur within 1-2 months after the lists were generated? A sentence at the end of the paragraph ending line 152 would be helpful, if word-count allows.

• We have included a sentence to specify the time of the tracing activities to “All tracing attempts occurred between 3 and 6 weeks after the lists were generated” See lines 174 - 175.

3) In response to item 6. Statistics have been added – the test name is ‘Pearson’s’ and it should be listed in the analysis section of the methods. The final p-value in Table 1 looked very surprising for these proportions and I think you will find it is 0.52.

• We have added the Pearson’s test to the data analysis paragraph. It reads, “We used Pearson’s chi square test to determine if there were any statistically significant differences in demographic and clinical characteristics and linkage outcomes between those who linked and had not linked to care.” See lines 197 - 1994) Minor errors or word changes to consider:

a. Line 49-51: Accurate data captured in ‘real time’ at PHC facilities will improve records and the resources to can be focused on truly lost to follow-up people.

• a) The entire abstract has been revised. The conclusion now contains more substantive recommendations, which has resulted in this sentence being removed. (See Lines 44-47)

b. Line 131 remove unnecessary comma and I suggest change ‘ago’ to ‘earlier’: generated a list of persons who had been diagnosed with TB three weeks earlier.

• b) Changed as per your suggestion (now line 153)

c. Line 224 add ‘were’: The majority (37/49;76%) of people with TB that we traced, found, and referred to care were linked to a TB treatment facility.

• c) Changed as per your suggestion (now line 249)

d. Line 227 add missing closing bracket: True loss to follow up was 30/107 (28%) among those not found and 12/49 (24%) among those found and referred to care (see Figure 3).

• d) Changed as per your suggestion (now line 252)

e. Line 241-242 add a space between semi-colon and % figures in brackets.

• e) Changed as per your suggestion (now line 255 - 256)

f. Line 245: I suggest you reference Figure 3 at end of the sentence.

• f) Changed as per your suggestion and revised the sentence for clarity (now line 269)

g. Reference 12 is not incomplete but it has ‘The’ immediately after the doi, which seems to be an error and should be deleted.

• g) Deleted as per your suggestion.

Reviewer 2: Comments

1) Lines 49-51: it appears there could be a missing word before “…the resources…”

Accurate data captured in ‘real time’ at 50 PHC facilities will improve records and the resources to be focused on truly lost to 51 follow-up people.

• The entire abstract has been revised. The conclusion now contains more substantive recommendations, which has resulted in this sentence being removed. See Lines 48- 52).

2) Lines 56-62. Suggest using the latest (2024) report.

• The Global TB Report (2024) has been used throughout, and all statistics quoted have been updated as per this latest report.

3) Lines 110-113 – suggest rewording.

People are typically first registered in a paper-based register and thereafter into the electronic TB treatment register at the facility, using the unique identification number “personal number/folder number”.

• Revised as suggested. Now lines 132-135

4) Lines 164 correct typo, ‘c’ missing in care.

• Changed, now line 173.

5) Lines 242-245:

check calculation or denominator. 1/30 is 0.3%.

Review last part of the sentence ie “…those found and referred to care with (17%) of death rate.”

• Changed, now line 268.

---

## [Editor Report · Decision Letter 2]

1 Apr 2025

PONE-D-24-44791R2Lessons from a systematic tracing process aimed to reduce initial loss to follow-up (ILTFU) among people diagnosed with tuberculosis (TB) in Cape Town, South AfricaPLOS ONE

Dear Dr. Vanqa,

Thank you for submitting your manuscript to PLOS ONE. After careful consideration, we feel that it has merit but does not fully meet PLOS ONE’s publication criteria as it currently stands. Therefore, we invite you to submit a revised version of the manuscript that addresses the points raised during the review process.

We look forward to receiving your revised manuscript.

Kind regards,

Daniel Chukwuemeka Ogbuabor, Ph.D., M.D.

Academic Editor

PLOS ONE

**Journal Requirements:**

**Additional Editor Comments:**

I must commend the authors for a significant improvement in the manuscript. We are almost there! Nonetheless, I have few vital points they need to address.

1. Introduction: The last paragraph of the introduction can improve if they author leverage some of the studies cited in the discussion and 3 other South African papers (see below after my comments) to rationalize the study and establish the knowledge gap on systematic tracing of ILTFU - yield, efforts required, and lessons learned.

2. Data analysis: Lines 200-202 indicates the authors "reflected on the activities and lessons learned from the systematic tracing." How was this reflection done? Who was involved, and for how long? Did you keep journals/ notes during the systematic tracing? It might be helpful to clarify the reflection process in the data collection and data analysis sub-sections of the methods.

3. Discussion: The authors should take a second look at the sentence in lines 295-296 and revise it to convey the intended meaning.

**Additional papers (refer to comment #1)**

ARTICLE

Scanty smears associated with initial loss to follow-up in South African tuberculosis patients

Claassens, M. M. ; Dunbar, R. ; Yang, B. ; Lombard, C. J.

The international journal of tuberculosis and lung disease, 2017-02, Vol.21 (2), p.196-201

ARTICLE

Initial loss to follow up among tuberculosis patients: the role of Ward-Based Outreach Teams and short message service

Mwansa-Kambafwile, Judith R. M ; Chasela, Charles ; Ismail, Nazir ; Menezes, Colin

BMC research notes, 2019-11, Vol.12 (1)

ARTICLE

Initial loss to follow up of tuberculosis patients in South Africa: perspectives of program managers

Mwansa-Kambafwile, Judith R M ; Jewett, Sara ; Chasela, Charles ; Ismail, Nazir ; Menezes, Colin

BMC public health, 2020-05, Vol.20 (1), p.622-622, Article 622

---

## [Author Response · Author response to Decision Letter 2]

16 Apr 2025

I must commend the authors for a significant improvement in the manuscript. We are almost there! Nonetheless, I have few vital points they need to address.

• Thank You.

1. Introduction: The last paragraph of the introduction can improve if they author leverage some of the studies cited in the discussion and 3 other South African papers (see below after my comments) to rationalize the study and establish the knowledge gap on systematic tracing of ILTFU - yield, efforts required, and lessons learned.

• We have revised the last paragraph of the introduction and cited additional papers, including the 3 named by the editor.

2. Data analysis: Lines 200-202 indicates the authors "reflected on the activities and lessons learned from the systematic tracing." How was this reflection done? Who was involved, and for how long? Did you keep journals/ notes during the systematic tracing? It might be helpful to clarify the reflection process in the data collection and data analysis sub-sections of the methods.

• We have added the detail details on our reflection process. See lines 194-200.

3. Discussion: The authors should take a second look at the sentence in lines 295-296 and revise it to convey the intended meaning.

• We have revised this sentence to clarify our intended meaning. It now reads, ‘Of those we traced and found at home, almost half had linked to care in the time between us checking the records at the health facility and us visiting them at home.’ Lines 303-304

---

## [Editor Report · Decision Letter 3]

17 Apr 2025

Lessons from a systematic tracing process aimed to reduce initial loss to follow-up (ILTFU) among people diagnosed with tuberculosis (TB) in Cape Town, South Africa

PONE-D-24-44791R3

Dear Nosivuyile Vanqa,

We’re pleased to inform you that your manuscript has been judged scientifically suitable for publication and will be formally accepted for publication once it meets all outstanding technical requirements.

Kind regards,

Daniel Chukwuemeka Ogbuabor, Ph.D., M.D.

Academic Editor

PLOS ONE
---

## [Editor Report · Acceptance letter]

PONE-D-24-44791R3

PLOS ONE

Dear Dr. Vanqa,

I'm pleased to inform you that your manuscript has been deemed suitable for publication in PLOS ONE. Congratulations! Your manuscript is now being handed over to our production team.

Kind regards,

on behalf of

Dr. Daniel Chukwuemeka Ogbuabor

Academic Editor

PLOS ONE